# Controlling Pre-trained Language Models
# for Grade-Specific Text Simplification

**Sweta Agrawal**
Department of Computer Science
University of Maryland
sweagraw@umd.edu

**Marine Carpuat**
Department of Computer Science
University of Maryland
marine@umd.edu

## Abstract

Text simplification (TS) systems rewrite text to make it more readable while preserving its content. However, what makes a text easy to read depends on the intended readers. Recent work has shown that pre-trained language models can simplify text using a wealth of techniques to control output simplicity, ranging from specifying only the desired reading grade level, to directly specifying low-level edit operations. Yet it remains unclear how to set these control parameters in practice. Existing approaches set them at the corpus level, disregarding the complexity of individual inputs and considering only one level of output complexity. In this work, we conduct an empirical study to understand how different control mechanisms impact the adequacy and simplicity of text simplification systems. Based on these insights, we introduce a simple method that predicts the edit operations required for simplifying a text for a specific grade level on an instance-per-instance basis. This approach improves the quality of the simplified outputs over corpus-level search-based heuristics.

## 1 Introduction

In the NLP task of text simplification, systems are asked to rewrite, restructure or modify an original text such that it improves the readability of the original text *for a target audience* while preserving its meaning. However, text can be simplified in many different ways and what makes a text simple to read depends on the reader. Replacing complex or specialized terms with simpler synonyms might help non-native speakers (Petersen and Ostendorf, 2007; Allen, 2009), restructuring text into short sentences with simple words might better match the literacy skills of children (Watanabe et al., 2009).

Acknowledging that text simplification is highly audience-centric (Stajner, 2021), recent work has focused on developing techniques to *control* the

**Original:** Paracho, the "guitar capital of Mexico," makes nearly 1 million classical guitars a year, many exported to the United States.

**Grade 5**: Paracho **is known as** the "guitar capital of Mexico." The town makes nearly 1 million classical guitars a year, with many exported to the United States.

`Word Length Ratio (W):` $1.19$

`Dependency Tree Depth Ratio (DTD):` $1.50$

**Grade 3**: Paracho **is known as** the "guitar capital of Mexico." The town makes **many guitars and sells some** in the United States.

`Word Length Ratio (W):` $0.96$

`Dependency Tree Depth Ratio (DTD):` $1.00$

Figure 1: Simplified texts can be obtained by either specifying the target audience (via grade level) or by using low-level control tokens to define the TS operation to be performed relative to the complex text (W, DTD).

degree of simplicity of the output at different levels. At a high level, one can simply specify the desired reading grade level of the output (Scarton and Specia, 2018; Kew and Ebling, 2022). At a low level, one can control complexity by describing the nature of simplification operations to be performed (Mallinson and Lapata, 2019; Martin et al., 2020). For example (Figure 1), one could obtain two distinct simplifications of the same inputs by indicating that they are intended for a grade 6 vs. grade 3 audience, or by specifying values for low-level control tokens such as the word length ratio (W) between the source and the target and the maximum dependency tree depth (DTD) ratio between the source and the target. For an original complex text at grade 8, when simplifying to grade 5, the low-level control values indicate a conservative rewrite, whereas, for grade 3, the properties encoded by the control tokens reflect a relatively more lexical and structural change.

While specifying a reading grade level might be more intuitive for lay users, it provides weaker control over the nature of simplification to be performed. On the other hand, controlling the outputs' simplicity by setting several low-level properties, such as the number of words or dependency tree depth, provides finer-grained control but can be cumbersome to set by readers, teachers, or other users. As a result, it remains unclear how to operationalize the control of text simplification in practice. Prior work sets low-level control values (length, degree of paraphrasing, lexical complexity, and syntactic complexity) at the corpus level by searching for control token values on a development set. This is done via maximizing a utility computed using an automatic evaluation metric, SARI, a metric designed to measure lexical simplicity (Xu et al., 2016). While this approach is appealing in its simplicity, it remains unclear whether this approach actually helps *control* complexity for individual inputs, as the control token values are always set at the corpus level.

This work presents a systematic empirical study of the impact of control tokens on the degree and quality of simplifications achieved at the instance level as measured by automatic text simplification metrics. Our empirical study shows that most corpus-level control tokens have an opposite impact on adequacy and simplicity when measured by BLEU and SARI respectively. As a result, selecting their values based on SARI alone yields simpler text at the cost of misrepresenting the original source content. To address this problem, we introduce simple models to predict what control tokens are needed for a given input text and a desired grade level, based on surface-form features extracted from the source text and the desired complexity level. We show that the predicted low-level control tokens improve text simplification on a controllable TS task compared to corpus-level search-based optimization.

## 2   Background on Controllable Text Simplification

While text simplification has been primarily framed as a task that rewrites complex text in simpler language in the NLP literature (Chandrasekar et al., 1996; Coster and Kauchak, 2011; Shardlow, 2014; Saggion, 2017; Zhang and Lapata, 2017), in practical applications, it is not sufficient to know that the output is simpler. Instead, it is necessary to target the complexity of the output language to a specific audience (Stajner, 2021). Controllable Text Simplification can be framed as a conditional language modeling task, where the source text $X$ is rewritten as an output $Y$ that presents attributes $V$ as scored by a model $P(Y|X, V)$ (Prabhumoye et al., 2020). In sequence-to-sequence models, techniques to control the properties $V$ during generation fall under two categories depending on whether they modify the training process (Sennrich et al., 2016; Holtzman et al., 2018; Dathathri et al., 2019; Li et al., 2022) as described below, or are supplied as constraints during inference (Hokamp and Liu, 2017; Ghazvininejad et al., 2017; Kumar et al., 2021). [1]

**Control Token Mechanisms**  A straightforward method to capture a target attribute, $V$, in text generation models is to represent it as a special token appended to the input sequence, $[V; X]$, which acts as a side constraint Sennrich et al. (2016). These constraints can be appended to the source or the target sequence.[2] The encoder learns a hidden representation for this token as for any other vocabulary token, and the decoder can attend to this representation to guide the generation of the output sequence. This simple strategy has been used to control second-person pronoun forms when translating into German (Sennrich et al., 2016), formality when translating to French (Niu et al., 2018), the target language in multilingual scenarios (Johnson et al., 2016) and to control style, content, and task-specific behavior for conditional language models (Keskar et al., 2019).

We provide an overview of the control tokens introduced in prior work for text simplification in Tables 1 and 2. Coarse-grained control over the degree and the nature of the simplification, e.g. via source and target grade levels is easier to interpret by end users (Table 1, `[1–6, 12]`), whereas controlling multiple low-level attributes (Table 1, `[7–11]`) that map text simplification operations to specific properties of the input and the output text can provide better control over the generated text. However, it is unclear how those low-level control values should be set during inference as these could vary significantly based on the **source**

---

[1]Please refer to Prabhumoye et al. (2020) for a full review on controllable text generation techniques.

[2]It is important to note that the term "control" used in this paper does not imply strict constraint enforcement on the model's output. Rather, the control tokens or side constraints merely serve as inputs that influence the model's behavior and encourage the generation of outputs with desired properties.

| Paper-ID | Paper | Control Tokens | How to Set? |
|---|---|---|---|
| [1] | Scarton and Specia (2018) | Target Grade and/or Operations | User-defined or Predicted |
| [2] | Nishihara et al. (2019) | | |
| [3] | Agrawal et al. (2021) | | |
| [4] | Yanamoto et al. (2022) | Target Grade (TG) | User-defined |
| [5] | Zetsu et al. (2022) | | |
| [6] | Agrawal and Carpuat (2022) | + Source Grade | |
| [7] | Martin et al. (2020) | ACCESS {C, L, WR, DTD} | |
| [8] | Sheang and Saggion (2021) | + W | Corpus-level Optimization with SARI |
| [9] | Martin et al. (2022) | − L + RL | |
| [10] | Maddela et al. (2021) | CC | Average over the training dataset |
| [11] | Qiao et al. (2022) | 10 Psycho-linguistic Features | Corpus-level Optimization with SARI |
| [12] | Kew and Ebling (2022) | $\lambda$ per grade-level | Hyperparameter Tuning with SARI |

Table 1: Control tokens define the nature and degree of simplifications either at a coarse-grained level such as specifying a target grade or via multiple low-level attributes like ACCESS. The control values are typically provided by the users or are set apriori during inference.

| ID | Name | Description |
|---|---|---|
| C | NbChars | character length ratio between source and target. |
| L | LevSim | character-level Levenshtein similarity (Levenshtein, 1966) between source and target. |
| WR | WordRank | ratio of log-ranks (inverse frequency order) between source and target. |
| DTD | DepTreeDepth | maximum depth of the dependency tree of the source divided by that of the target. |
| W | NbWords | word length ratio between source and target. |
| RL | Replace-only LevSim | character-level Levenshtein similarity only considering replace operations between source and target. |
| CC | Copy Control | percentage of copying between source and the target |

Table 2: Control Tokens introduced in prior work cover a wide range of TS edit operations.

**text** and **the degree of simplification** required. In all prior work (Martin et al., 2020, 2022; Sheang et al., 2022; Qiao et al., 2022), these values are set and evaluated at the corpus level. This is achieved by doing a hyperparameter search, optimizing for a single metric SARI on the entire validation set. SARI measures the lexical simplicity based on the n-grams kept, added, and deleted by the system relative to the source and the target sequence. We identify two key issues with this corpus-level search-based strategy for setting control values as described below:

**Input Agnostic Control**   Setting these control values at the corpus level disregards the nature and complexity of the original source text. It does not account for what can and should be simplified in a given input (Garbacea et al., 2021) and to what extent. We show that the control values are indeed dependent on all these factors as exhibited by a large variation observed in the values of the control tokens both at the corpus level (Figure 7) and individual target grade levels (Figure 9).

**Costly Hyperparameter Search**   Searching for control tokens value at the corpus-level is an expensive process. Martin et al. (2022) use the One-PlusOne optimizer with a budget of 64 evaluations using the NEVERGRAD library to set the 4 AC-CESS hyperparameters (up to 2 hours on a single GPU). Sheang et al. (2022) select the values that achieve the best SARI on the validation set with 500 runs. This takes $>= 3$ days when training the model takes only 10-15 hours. As these values are domain and corpus-specific, optimizing these values even at the corpus level for multiple datasets is computationally expensive.

We provide an analysis of the impact of these control values defined at the corpus level on the degree and nature of TS performed at the instance level in the next section.

## 3 How do Control Tokens Impact TS?

**Study Settings** We study the impact of setting the low-level control values at the *corpus level* on the *instance-level* simplification observed using automatic text simplification metrics. We conduct our analysis on the Newsela-grade dataset (Agrawal and Carpuat, 2019), which consists of news articles associated with multiple reference simplifications for diverse reading grade levels, and thus lets us analyze the degree and nature of simplification observed across inputs and target readability levels. This data is collected from Newsela,[3] an instructional content platform meant to help teachers prepare a curriculum that matches the language skills required at each grade level and has been used in prior work to benchmark controllable TS models (Scarton and Specia, 2018; Nishihara et al., 2019). It includes up to 4 text rewrites at various complexity levels (defined by U.S. reading grade levels 2-12) for an originally complex text. We use the control tokens defined in Sheang et al. (2022) (see Table 2 [1-5]) added to the source text as a side constraint in the format, W_{} C_{} L_{} WR_{} DTD_{} {Source_text}.

**Instance-Level Findings** Following prior work (Martin et al., 2020), we select the control tokens that maximize SARI on the Newsela-grade development set. We measure the complexity of the outputs generated and compare them with the complexity of the Newsela references by computing the Automatic Readability Index (Senter and Smith, 1967, ARI, Refer Equation 1).

As can be seen in Figure 2, control tokens set at the corpus level using SARI tend to over or under-simplify individual input instances. When the reference distribution exhibits diversity in the degree of the simplification performed, setting corpus-level values is sub-optimal: outputs are frequently over- or under-simplified as illustrated by the difference in the reference and predicted grade levels.

**Corpus-Level Findings** Figure 3 shows the correlation between 100 sets of control values set at the corpus level and automatic TS metrics computed using the outputs generated: SARI, BLEU, and FR

[3]newsela.com

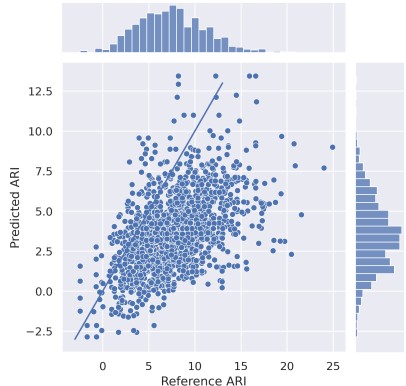

Figure 2: ARI accuracy on the Newsela-grade Development Set: 12%. Setting corpus-level control values results in over or under-simplification.

(Flesch Reading Ease): Most control tokens have an opposite impact on SARI and BLEU, except the character length ratio, (C). This suggests that setting their values by optimizing for SARI alone at the corpus level can be misleading, as a high SARI score can be achieved at the expense of a lower adequacy score. These findings are consistent with the observation of Schwarzer and Kauchak (2018) who note a similar negative correlation between human judgments of simplicity and adequacy and caution that: "improvement in one metric and not the other may be due to this inverse relationship rather than actual system performance". These results lead us to concur with the recommendation of Alva-Manchego et al. (2021), which advocates for always augmenting SARI with an adequacy metric for text simplification evaluation.

Overall, this analysis highlights important limitations of the simplification abilities provided by setting control tokens based on optimizing SARI at the corpus level. We propose instead a simple method to predict these values based on each input instance and the desired output complexity.

## 4 Grade-Specific Text Simplification with Instance-Level Control

Since the simplification for a given instance should depend on the original source text, its complexity, and the desired target complexity, we introduce a Control Predictor module (CP) that predicts a vector of control token values $V$ for each input $X$ at inference time. Figure 4 shows the overall inference pipeline for generating the simplified text using the control token values predicted using $CP$.

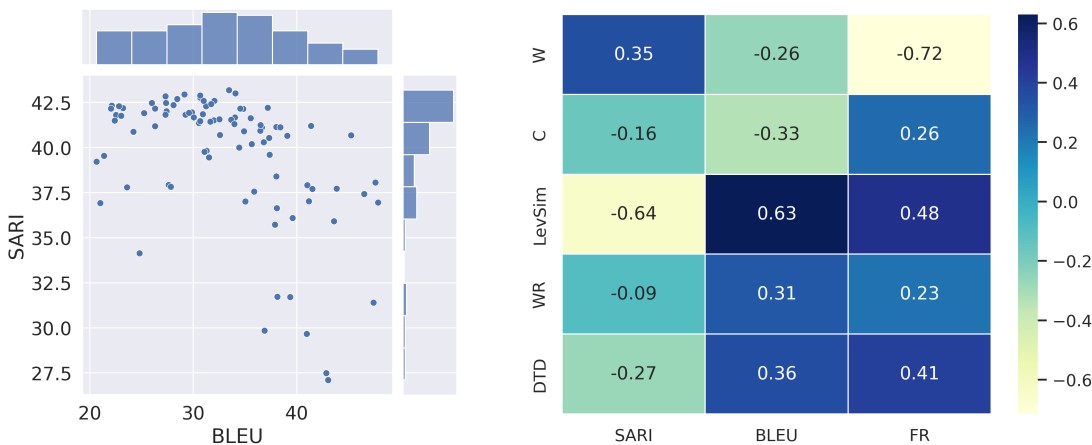

Figure 3: Adequacy-Simplicity Tradeoff on the Newsela-Grade development set when using 100 different control tokens set at the corpus level: Most control tokens have an opposite impact on BLEU and SARI, suggesting that setting their values on SARI alone can be misleading.

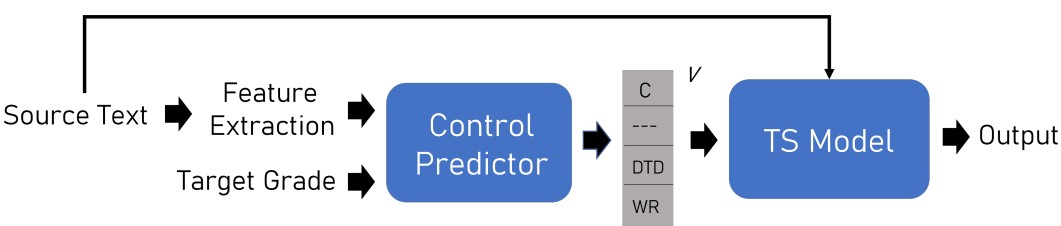

Figure 4: At inference time, low-level control tokens are first estimated via the control predictor using the source text and a user-defined target grade level. The low-level tokens are then fed as input to the TS generation model to produce a simplified output.

**Predicting Control Tokens** We thus directly train a Control Predictor (CP($\theta$): $X \rightarrow V$) to predict the control vector given features extracted from an input text and the input and output grade levels. Let $\{x_i, y_i\} \in D$ represent a complex-simple pair, and the ACCESS controls associated with this pair be $V_i = \{W_i, C_i, L_i, WR_i, DTD_i\}$. We propose both single and multi-output regression solutions for predicting $V$ as described below:

1. **CP-Single:** The model is trained to predict the individual control tokens, resulting in one model per control value.

2. **CP-Multi:** The model is trained to optimize the mean RMSE error over all the control dimensions.

We train a simple feature-based Gradient Boosting Decision Trees classifier [4] to predict the control values, $V$ using the CatBoost library using several surface-form features extracted from the source text as described below:

1. Number of Words

[4] https://catboost.ai/en/docs/

2. Number of Characters
3. Maximum Dependency Tree Depth
4. Word Rank
5. Mean Age of Acquisition (Schumacher et al., 2016)

We incorporate the source and target grade levels as attributes to accommodate the differences in control token values resulting from the level of simplification needed.

**TS Model Training** Given a source text ($x$) and a control vector $v$, the controllable TS model $P(y|x, v)$, is trained to generate a simplified output ($y$) that conforms to $v$ in a supervised fashion, by setting $v$ to oracle values derived from the reference and optimizing the cross-entropy loss on the training data.

## 5 Experimental Settings

**Data** We use the Newsela-grade dataset (Agrawal and Carpuat, 2019) with 470k/2k/19k samples for training, development and test sets respectively.

**Metrics** We automatically evaluate the truecased detokenized system outputs using:

1. **SARI** (Xu et al., 2016), which measures the lexical simplicity based on the n-grams kept, added, and deleted by the system relative to the source and the target. [5]

2. **BERTSCORE** (Zhang et al.) for assessing the output quality and meaning preservation

3. **ARI-Accuracy** (Heilman et al., 2008) that represents the percentage of sentences where the system outputs' ARI grade level is within 1 grade of the reference text, where ARI is computed as:

$$ARI = 4.71(\frac{chars}{words}) + 0.5(\frac{words}{sents}) - 21.43. \tag{1}$$

4. **%Unchanged Outputs (U)** The percentage of outputs that are unchanged from the source (i.e., exact copies).

We evaluate the fit of the control predictor in predicting $V$ using **RMSE** and **Pearson Correlation** with the gold ACCESS values.

**Model Configuration** We finetune the T5-base model following Sheang et al. (2022) with default parameters from the Transformers library except for a batch size of 6, maximum length of 256, learning rate of 3e-4, weight decay of 0.1, Adam epsilon of 1e-8, 5 warm-up steps, and 5 epochs. For generation, we use a beam size of 8. We train all our models on one GeForce RTX 2080Ti GPUs. Training takes 7-8 hours to converge.

We use a learning rate of 0.1 and a tree depth of 6 for training all the control predictor models which takes approximately 5-10 minutes.

**Controllable TS Variants** We compare the prediction-based TS models above with two variants:

- GRADE TOKENS: a model that uses high-level control token values, i.e. the source grade (SG) and the target grade (TG) levels when finetuning the generation model (Scarton and Specia, 2018).

- AVG-GRADE: a simple approach that sets control values with the average of the values observed for the source-target grade pair.

**Controllable TS Baselines** We compare our approach with the corpus-level hyperparameter search strategy (CORPUS-LEVEL) used in prior work that selects the best low-level control values based on SARI only (Martin et al., 2020).

**Source Grade at Inference** While the desired target grade level is known during inference, we automatically predict the grade level of each source sentence using the ARI score in all the settings.

# 6 Results

We first discuss the accuracy of the Control predictor in estimating the ACCESS control values on the Newsela-Grade dataset and then show the impact of using the predicted control tokens as constraints towards controlling the degree of simplification in the generated outputs.

## 6.1 Intrinsic Evaluation of Control Predictor

| CONTROL | CORRELATION(↑) | | | | | RMSE (↓) |
|---|---|---|---|---|---|---|
| | W | C | LevSim | WR | DTD | |
| CP-SINGLE | 0.405 | 0.407 | 0.567 | **0.398** | 0.567 | 0.197 |
| CP-MULTI | **0.420** | **0.422** | **0.568** | 0.393 | **0.570** | **0.196** |

Table 3: `CP-Multi` improves correlation (averaged over 10 runs) with ground truth low-level control token values (`W`, `C`) on Newsela-grade development set over `CP-Single`.

Table 3 shows the correlation and RMSE of predicted values with gold low-level control tokens. Training the model to jointly predict all values, $V$ improves correlation ($+0.015$) for `W`, `C` over training independent models (`CP-Single`) for individual control tokens. We show that this can be attributed to the correlation amongst the target control values in Figure 5. Both (`W`, `C`) exhibit moderate correlation with `DTD`. There is a drop in correlation for `WR` when training a joint model which is expected as `WR` is a proxy for lexical complexity and is the most independent control token.

The correlation scores for the control tokens (`W`, `C`, `DTD`, `LevSim`, `WR`) range from $-0.33$ (`S_W`, `W`) to $0.49$ (`TG`, `LevSim`). The moderate correlation between the source features (prepended with S) and the low-level tokens suggests that the source text influences the nature of simplification that can be performed. Additionally, `LevSim` controls the degree of simplification as suggested by its moderate-high correlation with

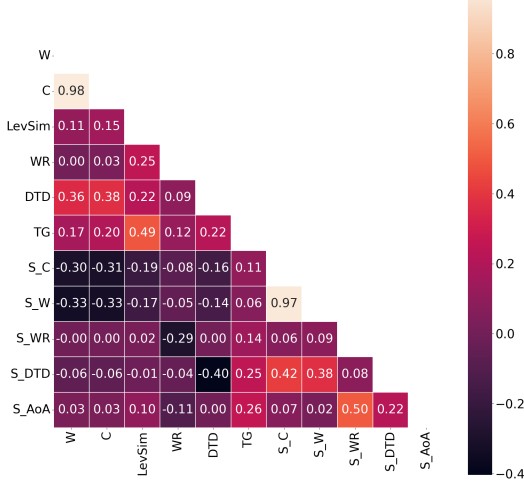

Figure 5: Correlation between source features and control token values on the Newsela-Grade training set.

the target grade level and can be considered the most prominent token for balancing the adequacy-simplicity tradeoff.

## 6.2 Overall Grade-Specific TS Results

We show how the different control tokens as side constraints influence the degree and the nature of simplification in the generated outputs in Table 4.

| CONTROL | BERTSCORE | SARI | %ACC | %U |
|---|---|---|---|---|
| *Low-level* | | | | |
| CORPUS-LEVEL | 0.922 | 42.19 | 3.1 | 0 |
| AVG-GRADE | 0.945 | 44.09 | 32.7 | 0 |
| CP-SINGLE | 0.946 | 45.54 | 36.3 | 2 |
| CP-MULTI | 0.946 | **45.65** | 36.3 | 3 |
| *High-level* | | | | |
| GRADE TOKENS | **0.955** | 43.02 | **39.8** | 34 |
| ORACLE | 0.955 | 53.53 | 56.7 | 12 |

Table 4: Results on the Newsela-grade dataset: using source-informed tokens (CP-∗) significantly improves SARI over alternative control mechanisms. All differences are significant except the difference between CP-Single and CP-Multi with p-value of 0.00.

**Setting corpus-level control for grade-specific TS is suboptimal.** Optimizing SARI alone for selecting the low-level control tokens and setting corpus-level control values is suboptimal for matching the desired complexity level. This is indicated by the low ARI accuracy of only 3.1%.

**Predictor-based instance-level control outperforms grade or corpus-level control.** Predictor-

based models (`CP-Single, CP-Multi`) that set control tokens for each instance based on source features improve simplicity scores compared to using `Avg-Grade`, which only uses grade information to set control values. These models show improvements in SARI (+1.4-1.5) and ARI (+3.6%) scores, highlighting the importance of setting control tokens at the instance level rather than relying solely on just the grade information. Furthermore, setting control tokens based on the average values observed for a given source-target grade pair, i.e., `Avg-Grade` significantly improves both BERTSCORE and ARI-based metrics across the board compared to the `Corpus-level` approach.

**Grade-level (high) and operation-specific (low) control tokens exhibit different adequacy and simplicity tradeoffs.** Low-level control tokens offer more precise control over the simplicity of outputs, resulting in improved SARI scores by at least 2 points compared to `Grade Tokens`. However, this advantage comes at the cost of lower adequacy (BERTScore) and control over desired complexity (ARI Accuracy). The models trained with low-level control values exhibit lower grade accuracy scores partly due to the limited representation of the need for text simplification (Garbacea et al., 2021) during the generation process as suggested by a lower percentage of exact copies in the output compared to `Grade Tokens` (Exact copies in references: 12%). On the subset of the test set with no exact matches between the source and the reference text, `Grade Tokens` and `CP-Multi` receive ARI accuracy of 34.2 and 34.0 respectively. Furthermore, we hypothesize that the models trained with low-level control exhibit low meaning preservation because none of the control tokens directly encourage content addition during text simplification. And, while the model learns to perform an appropriate content deletion, it does not generate a fitting substitution or addition as required to preserve the meaning of the original source text. We show a detailed operation-specific analysis in the following section.

## 6.3 Impact of Control Tokens on TS Edit Operations

**Predicting control tokens for individual instances improves coverage over the range of control values exhibited by the oracle.** We show the distribution of control values observed by differ-

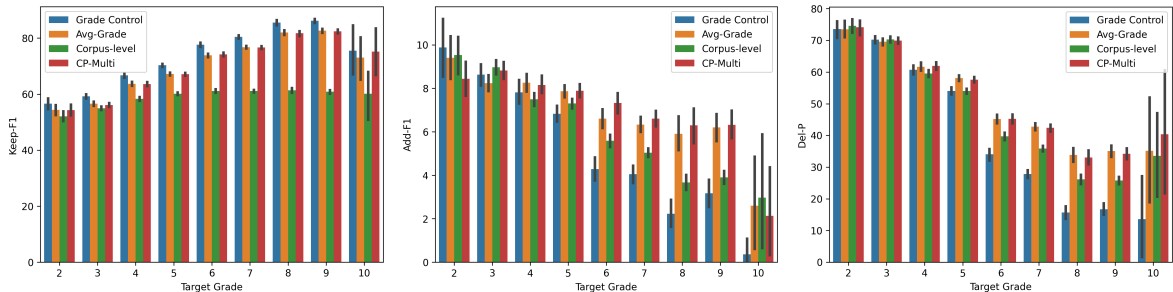

Figure 6: Edit Operations by Target Grade Levels: CP-Single performs correct and diverse edits as suggested by the high Add-F1 and Del-P scores for all target grade levels > 4.

ent ways of setting the low-level control values in Figure 7. The high variance in the range of oracle values confirms that the control tokens vary based on the source texts' complexity and desired output complexity. Where `Corpus-level` fails to even match the mean distribution, `CP-Multi` is able to cover a wider range of control values. We show that this trend holds across all target grade levels in the Appendix Figure 9.

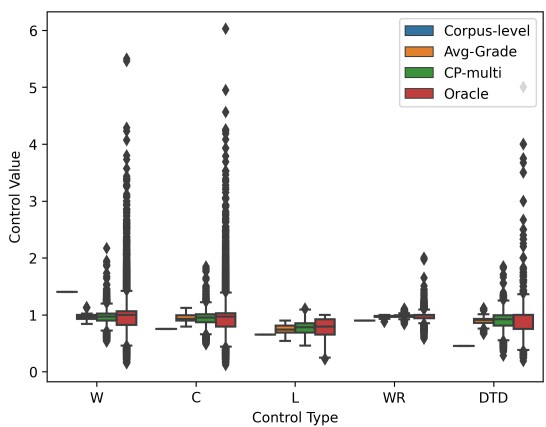

Figure 7: Distribution of control values for different control mechanisms: `CP-multi` provides a broader coverage of control values as observed in the oracle distribution over `Corpus-level` and `Avg-Grade`.

**Simplified outputs generated using predicted control tokens exhibit diverse edit operations.** Figure 6 shows the distribution of the KEEP-F1, DEL-P, and ADD-F1 scores by target grade level for the models trained with different control types, where ADD-F1 computes the F1 score for the n-grams that are added to the system output relative to the source and the reference text. The model's deletion capability is measured by the F1 score for n-grams that are kept (KEEP-F1) and the precision of deletion operation (DEL-P) with respect to the source and the reference.

`CP-Multi` consistently achieves better or competitive DEL-P across all target grade levels over alternative control mechanisms, suggesting that setting control values informed by both the source and desired complexity level improves the model's ability to appropriately delete redundant information. The former also generally improves ADD-F1 scores, highlighting that the model also appropriately performs lexical substitution or content addition as required across different grade levels (except grades 2 and 10). Moreover, low-level control tokens (`CP-Multi`, `Avg-Grade`) exhibit more diverse and correct modifications compared to high-level control (`Grade Tokens`), as evident from their better ADD-F1 and DEL-P scores for grade levels > 3, where the latter prioritizes meaning preservation (high KEEP-F1).

## 7 Conclusion

We present a systematic analysis of the impact of control tokens set at the corpus level on the degree and quality of simplification achieved by controllable text simplification models at the instance level. Our findings show that control tokens exhibit an opposite correlation with adequacy and simplicity. Hence, selecting their values at the corpus level based on SARI alone leads to over or under-simplifying individual instances. This motivates a new approach to set low-level control tokens during inference by predicting them given a source text and desired target grade level. We show that this approach is effective at improving the quality and controlling the degree of simplification in generated outputs based on automatic evaluation. Furthermore, predicted low-level control tokens yield more diverse edit operations than alternative ways of setting control on the Newsela-grade dataset.

Our proposed simple solutions improve the inference capability of the controllable TS model

for grade-specific TS and reduce the gap with the oracle over a corpus-level baseline approach. However, more sophisticated techniques can benefit the design and prediction of low-level control values and their usage during inference which we leave to future work.

## Limitations

We note a few limitations of our work. While our proposed strategies are simple and improve the controllability over the generated simplified texts during inference, the models trained with low-level control tokens struggle to identify when a text needs to be simplified compared to the model that uses high-level weak supervision. These results open space for further research in designing end-to-end controllable TS models that are able to take advantage of both high and low-level control tokens for controlling both the degree and the nature of simplification.

Our work is also limited to one dataset and one language (English) and hence studies the mapping between U.S grade level to low-level edit operations. It remains an open question to study how the control predictor would generalize in other settings, datasets, and language pairs.

## Ethics Statement

This work is conducted in full awareness of and in line with the ACL Ethics Policy. Models, datasets, and evaluation methodologies used are detailed in Section 5. The Newsela dataset was used with permission and appropriate access rights and licenses. And, we ground our claims by conducting a thorough evaluation and analysis of the outputs generated by the proposed systems (Section 6).

We note that while text simplification systems are designed with the intention of assisting users in better comprehending complex texts, the potential errors introduced by these systems and ambiguous interpretations of simplified text can cause harm to the reader and other stakeholders. The very nature of simplification involves content removal and rephrasing complex concepts, which can sometimes result in oversimplification or loss of critical nuances. As a consequence, users relying solely on simplified texts may develop an incomplete or inaccurate understanding of the subject matter, leading to potential misconceptions or misinterpretations.

## Acknowledgments

We thank Eleftheria Briakou, Neha Srikanth, the members of the CLIP lab at UMD, and the anonymous EMNLP reviewers for their helpful and constructive comments. This research is supported in part by the Office of the Director of National Intelligence (ODNI), Intelligence Advanced Research Projects Activity (IARPA), via the HIATUS Program contract 2022-22072200006, by NSF grant 2147292, and by funding from Adobe Research. The views and conclusions contained herein are those of the authors and should not be interpreted as necessarily representing the official policies, either expressed or implied, of ODNI, IARPA, or the U.S. Government. The U.S. Government is authorized to reproduce and distribute reprints for governmental purposes notwithstanding any copyright annotation therein.

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

# A    Impact of Training Data Size on Control Predictor

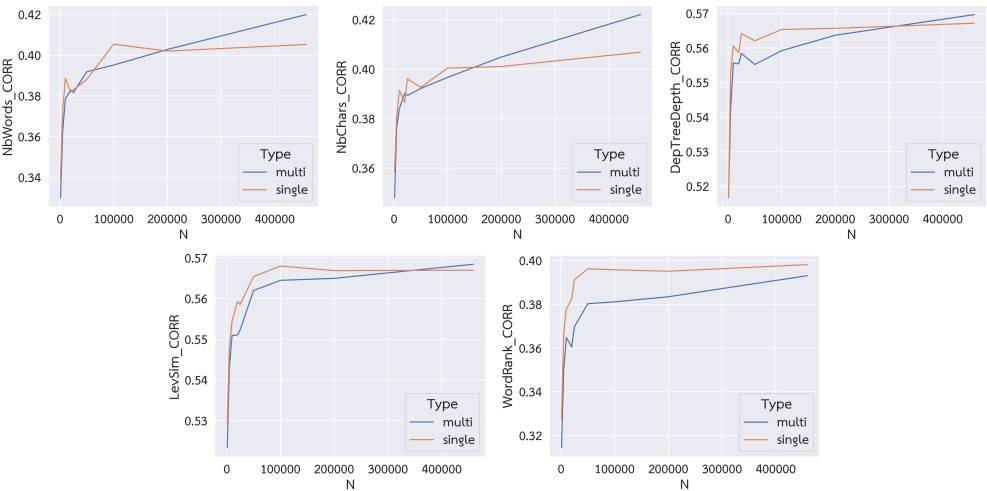

Figure 8: Correlation scores for all low-level control tokens with varying training dataset sizes.

We vary the size of the dataset used to train the single and multi-regressor control predictors and show the correlation for all the control values, $V$, in Figure 8. While correlation for CP-SINGLE saturates with $100 - 150K$ instances, CP-MULTI is able to take advantage of correlation amongst tokens and additional training dataset to further improve the prediction of ACCESS control tokens.

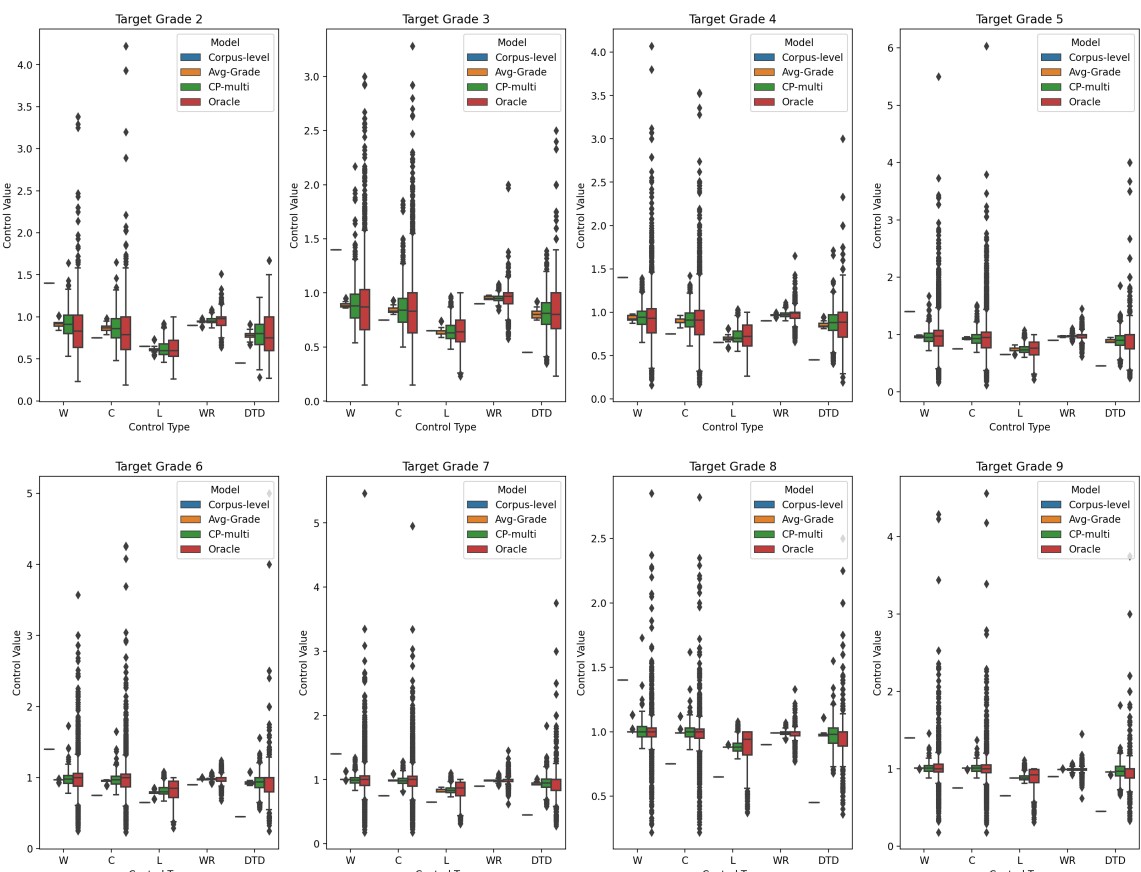

Figure 9: Distribution of control token values for different model variants by Target Grade level.