# OpenReview forum: "Controlling Pre-trained Language Models for Grade-Specific Text Simplification"
_EMNLP/2023/Conference — EMNLP 2023 Main_

### Official Review · Reviewer_Sk1b · 2023-08-05

**Soundness:** 2

**Excitement:**

3: Ambivalent: It has merits (e.g., it reports state-of-the-art results, the idea is nice), but there are key weaknesses (e.g., it describes incremental work), and it can significantly benefit from another round of revision. However, I won't object to accepting it if my co-reviewers champion it.

**Paper Topic And Main Contributions:**

This paper first conducted an empirical survey on the impacts of different control mechanisms on the simplicity of texts. They then provide an approach to first predict the simplification edit operation and then do a second stage rewriting of the texts.

**Questions For The Authors:**

- The high BERTScore suggests that the outputs are nearly the same as the inputs, right?
- How do you compute the ARI accuracy and how can we interpret the 30%+ accuracy and the gaps?

**Reasons To Accept:**

- I appreciate the authors for the good motivation to better understand the effects of different control mechanisms and design the two staged models to infer control tokens first.

**Reasons To Reject:**

- There is a lack of examples to demonstrate the effects of different systems. While claiming that the Automatic metrics are not good or even negatively correlated with the simplification quality (i.e. SARI), I would like to see the human evaluations or some real examples.
- The innovation of the proposed tasks seems to replace the Control-token values with a new set, which is incremental and the results in Table 4 are super close to each other.

**Reproducibility:**

3: Could reproduce the results with some difficulty. The settings of parameters are underspecified or subjectively determined; the training/evaluation data are not widely available.

**Reviewer Confidence:**

4: Quite sure. I tried to check the important points carefully. It's unlikely, though conceivable, that I missed something that should affect my ratings.

---

> ### Author Rebuttal · Authors · 2023-08-29
>
> Thank you for your careful reading of our paper and your acknowledgment of the motivation and design of our study. We provide our response to your questions below and hope it addresses your concerns.
>
> Q1 High BERTScore: A high BERTScore indicates that the outputs have the same meaning as the inputs. While outputs that barely change from the input can get high BERTScores, Figure 6 (% tokens that are kept the same, deleted, or added by each of these systems) shows that this is not the case here. We see the outputs are notably different from the original text across different systems and grade levels while preserving meaning based on the BERTScore.
>
> Q2 Interpretation of ARI Accuracy: We follow Heilman et al. (2008) and compute the accuracy as the % output instances that are within 1-grade level of the desired target grade level.  The higher the accuracy, the output better matches the desired complexity according to an automatic readability index. We report the ARI Accuracy for all models and comparisons in Table 6 and also study how these scores vary across grade differences (i.e., the difference between the source grade and the intended target grade). As discussed in L442-446 our findings show that models that copy text or are more meaning-preserving (e.g. Grade Tokens) can achieve a high or perfect score when the source matches the reference, i.e. when grade difference is low and no simplification is required. Hence a more fine-grained analysis by bucketing the instances based on target grade levels or desired level of simplification can help interpret the gap. We are happy to include such an analysis in the next version of the paper for all grade differences.
>
> Comparison between different systems and human study: We show that our proposed control mechanisms outperform the previous corpus-level approach and other baselines for grade-specific TS via a thorough automatic evaluation in section 6.2.  The numbers in Table 4 while close, are statistically significant with a p-value of 0.00 on a test set of size 20k. While we do not show individual qualitative examples due to the complexity of evaluating fine-grained differences between different systems at the instance level across different target grade levels, we measure the impact of the different control mechanisms for generating the correct TS edit operation (keep/add/delete) for individual target grade levels in Figure 6, where we show that our proposed mechanism, CP-Multi achieves better or competitive precision of deletion operation as well as improved F1 score for addition operation across target grade levels over baselines.  Again, we acknowledge that a human evaluation with experts could be interesting, however, designing an appropriate evaluation framework or user study for audience-specific TS is an open question in itself which deserves a separate paper.
>
> Use of Automatic Metrics: We never claim in the paper that automatic metrics are not good. Instead, we question the way of setting control tokens at the corpus level using just a single evaluation metric, which can result in over or under-simplified outputs (Figure 2). We show that most control tokens show an opposite impact on adequacy-based metric BLEU and simplicity metric SARI in Figure 3, which implies that one can improve SARI by setting low values of these control tokens but significantly compromise the adequacy of the generated outputs. Hence, we argue that these values should be set at the instance level based on the source text and the desired target complexity. We also refer the reader to the examples in Table 4 of the original paper [1] where the authors note that a low character-level Levenshtein similarity can result in hallucinated content in the simplified text for some inputs.
>
> Incremental Innovation: As text simplification efforts focus on tailoring content to specific audiences, adaptable control mechanisms have been developed to address various audience needs. However, there's a gap between how these mechanisms are designed and their practical use during inference. Setting these control values globally via corpus-level optimization can lead to both overly simplistic or complex outputs and is also computationally sub-optimal (Sections 2 & 3). To address this, we propose simple but effective methods that use the source text and desired grade level to improve the quality and control on a controllable grade-specific TS dataset, maintaining interpretability and output variation (Section 6). With a small supervised dataset, the proposed control predictor can be trained and adapted for other contexts and audiences. Our novelty thus lies in identifying and developing a method to enhance controllable TS through interpretable and contextually relevant control mechanisms.

---

### Official Review · Reviewer_jeYZ · 2023-08-11

**Soundness:** 3

**Excitement:**

3: Ambivalent: It has merits (e.g., it reports state-of-the-art results, the idea is nice), but there are key weaknesses (e.g., it describes incremental work), and it can significantly benefit from another round of revision. However, I won't object to accepting it if my co-reviewers champion it.

**Paper Topic And Main Contributions:**

The paper conducts an empirical study to evaluate different control mechanisms for text simplification. Also, it introduces a new text simplification method using a feature-based Gradient Boosting Decision Trees on an instance-per-instance basis.

**Questions For The Authors:**

a. Have you tested with other Control Predictor features? How relevant is each feature?

**Reasons To Accept:**

- The proposed model is simple and achieves good results.
- The work provides an extensive evaluation of previous methods.

**Reasons To Reject:**

- It would be enriching to have an explanation of how the Control Predictor features were selected.

**Reproducibility:**

4: Could mostly reproduce the results, but there may be some variation because of sample variance or minor variations in their interpretation of the protocol or method.

**Reviewer Confidence:**

3: Pretty sure, but there's a chance I missed something. Although I have a good feel for this area in general, I did not carefully check the paper's details, e.g., the math, experimental design, or novelty.

**Typos Grammar Style And Presentation Improvements:**

- Define the abbreviation TS (text simplification) before start using it.
- Line 50: grade 6 -> grade 5, as Figure 1 indicates.
- Line 118: remove "for".
- Line 183: upto -> up to.
- Lines 340-342: remove the sentence "The learning rate and [...] models.", it is repeated in the next paragraph.

---

> ### Author Rebuttal · Authors · 2023-08-29
>
> We thank the reviewer for their careful reading and valuable suggestions in improving the presentation of the paper and will incorporate those in the camera ready version of the paper. We also appreciate that the reviewer found our evaluation extensive with previous methods.
>
> Selection of Control Features: The character and word length of the source, the source dependency tree depth and the word ratio are features that directly impact the control token values by the definition (Table 2). These are also the features that have been used to determine the complexity of a text in prior work. We further included the age of acquisition, as this feature was shown to have high correlation with sentence level difficulty [1]. We do not examine the impact of individual features via a leave-one-out ablation, but we support their inclusion in a multi-regression framework via a correlation analysis, as depicted in Figure 5, along with a comprehensive discussion in lines 376-399.
>
> [1] Elliot Schumacher, Maxine Eskenazi, Gwen Frishkoff, and Kevyn Collins-Thompson. 2016. Predicting the Relative Difficulty of Single Sentences With and Without Surrounding Context. In Proceedings of the 2016 Conference on Empirical Methods in Natural Language Processing, pages 1871–1881, Austin, Texas.

---

### Official Review · Reviewer_ctbg · 2023-08-11

**Soundness:** 4

**Excitement:**

4: Strong: This paper deepens the understanding of some phenomenon or lowers the barriers to an existing research direction.

**Paper Topic And Main Contributions:**

The task in this paper is text simplification. The authors present a systematic analysis of the usage of setting control tokens at a corpus level vs at a low level. They show that setting low level control tokens during inference, on an instance level based on the source text and target grade level is better at simplification, with a greater level of control and quality on diverse edits as opposed to corpus level tokens commonly used.

**Questions For The Authors:**

1. The authors use T5 model to finetune and study the low level control tokens. I’m curious if the authors experimented with any other models, and if they are able to achieve this level of control using large language model prompting. Note that not using LLMs is not a criticism, I am just curious.

2. Why did you choose not to run a human study? Since the automatic metrics do not provide a full picture and end user usage might be different.


**Reasons To Accept:**

The authors provide an in depth analysis of the usage of controllable text simplification using simple low level tokens, which can be directly beneficial with a target grade token to help people with varying literacy levels to understand and comprehend text. I believe this work is very focused.


**Reasons To Reject:**

The authors studied one dataset, however it would be interesting to see how a domain switch would affect the usability of controllable text simplification. Further, they used only automatic metrics to measure quality with no human evaluation - which may or may not correlate with the actual downstream users’ preference.


**Reproducibility:**

3: Could reproduce the results with some difficulty. The settings of parameters are underspecified or subjectively determined; the training/evaluation data are not widely available.

**Reviewer Confidence:**

4: Quite sure. I tried to check the important points carefully. It's unlikely, though conceivable, that I missed something that should affect my ratings.

---

> ### Author Rebuttal · Authors · 2023-08-29
>
> Thank you for your feedback on our paper. We are glad that you found our work focused and potentially beneficial to users. We provide responses to your questions below:
>
> Q1 Controllable TS with LLMs: We did not experiment with other models. Our preliminary experiments with large language models showed limited control over the degree of simplification with prompting. We note that the choice of prompts could also have impacted our findings, as LLMs are shown to be sensitive to the prompt design. We leave the full exploration of LLMs (as well as strategies to design prompts to enable controlled generation) to future work.
>
> Q2 Human Study: We followed automatic evaluation of controllable TS following best practices from prior work (cite). Our automatic evaluation (Section 6.2) and a thorough ablation (Sections 3, 6.3) demonstrate a clear advantage of aligning control tokens with Newsela grade levels over the corpus-level SARI-based optimization (Table 4, Section 6.2). We agree that a human study would be very interesting, however, we left that endeavor to future work, as how to soundly design such an evaluation for audience-specific text simplification is an open question in itself which deserves a separate paper.
>
> Domain Switch: Indeed, we did not explicitly test for the impact of domain switch on output quality. Our training and evaluation dataset are sourced from Newsela which includes texts from a range of topics. Like any other text generation tasks, we expect the output quality to drop in unrelated new domains. However, as our proposed control predictor (Section 4) is trained on surface form features like word and character length, we expect the predictor to be more robust to domain switch.

---

### Official Review · Reviewer_ydY4 · 2023-08-12

**Soundness:** 4

**Excitement:**

4: Strong: This paper deepens the understanding of some phenomenon or lowers the barriers to an existing research direction.

**Paper Topic And Main Contributions:**

In this paper authors analyze impact of various control tokens on creating text simplification at instance level with controlable framework, the work also analyzes the limitations of applying control tokens at corpus level. The proposed framework will give user the flexibility to select grade of the output and input along with the input text to get the desired simplified text.

**Reasons To Accept:**

Analysis in the paper is thorough in comparing various methods and demonstrates impact of the proposed framework on the metrics along with various scenarios and how the proposed framework performs.

**Reasons To Reject:**

It would be helpful to add definitions for critical keywords used in the paper like Grades of the simplification, also adding more details on various control tokens instead of just providing the references in table 1.

**Reproducibility:**

3: Could reproduce the results with some difficulty. The settings of parameters are underspecified or subjectively determined; the training/evaluation data are not widely available.

**Reviewer Confidence:**

4: Quite sure. I tried to check the important points carefully. It's unlikely, though conceivable, that I missed something that should affect my ratings.

---

> ### Author Rebuttal · Authors · 2023-08-29
>
> Thank you for your feedback. We are pleased that the analysis presented in the paper on comparing various control mechanism and their impact on text quality was acknowledged to be thorough.
>
> We already provide the definitions of the control tokens in Table 2 and illustrate a subset of them in Figure 1. We can illustrate them further by adding examples to illustrate the impact of different control mechanisms on the output in the camera-ready version of the paper.

---

### Meta-Review · Area_Chair_cCFf · 2023-09-16

**Recommendation:** 4

**Metareview:**

This paper presents a new method that predicts the edit operations required to simplify a text for a specific grade level on an instance-per-instance basis, which improves the quality of the simplified outputs over existing methods.
There are concerns that the proposed method has only been verified on one dataset and there is no human evaluation.

---

### Decision · Program_Chairs · 2023-10-07

**Decision:**

Accept-Main

**Comment:**

This paper presents a new method that predicts the edit operations required to simplify a text for a specific grade level on an instance-per-instance basis, which improves the quality of the simplified outputs over existing methods.
There are concerns that the proposed method has only been verified on one dataset and there is no human evaluation.